# Short-Chained Alcohols Make Membrane Surfaces Conducive for Melittin Action: Implication for the Physiological Role of Alcohols in Cells

**DOI:** 10.3390/cells11121928

**Published:** 2022-06-15

**Authors:** Haoyu Wang, Hao Qin, Győző Garab, Edward S. Gasanoff

**Affiliations:** 1STEM (Science, Technology, Engineering and Mathematics) Program, Science Department, Chaoyang KaiWen Academy, Beijing 100018, China; 2008230017@cy.kaiwenacademy.cn (H.W.); hao.qin@cy.kaiwenacademy.cn (H.Q.); 2Biological Research Centre, Eötvös Loránd Research Network, Temesvári krt. 62, H-6726 Szeged, Hungary; 3Department of Physics, Faculty of Science, University of Ostrava, 710 00 Ostrava, Czech Republic; 4Belozersky Institute for Physico-Chemical Biology, Lomonosov Moscow State University, 119991 Moscow, Russia

**Keywords:** alcohols, melittin, heart rate, mitochondrial ATP production, non-bilayer structures, ERP of spin probes, ^1^H-NMR

## Abstract

Alcohols are a part of cellular metabolism, but their physiological roles are not well understood. We investigated the effects of short-chain alcohols on *Daphnia pulex* and model membranes mimicking the lipid composition of eukaryotic inner mitochondrial membranes. We also studied the synergistic effects of alcohols with the bee venom membrane-active peptide, melittin, which is structurally similar to endogenous membrane-active peptides. The alcohols, from ethanol to octanol, gradually decreased the heart rate and the mitochondrial ATP synthesis of daphnia; in contrast, in combination with melittin, which exerted no sizeable effect, they gradually increased both the heart rate and the ATP synthesis. Lipid packing and the order parameter of oriented films, monitored by EPR spectroscopy of the spin-labeled probe 5-doxylstrearic acid, revealed gradual alcohol-assisted bilayer to non-bilayer transitions in the presence of melittin; further, while the alcohols decreased, in combination with melittin they increased the order parameter of the film, which is attributed to the alcohol-facilitated association of melittin with the membrane. A ^1^H-NMR spectroscopy of the liposomes confirmed the enhanced induction of a non-bilayer lipid phase that formed around the melittin, without the permeabilization of the liposomal membrane. Our data suggest that short-chain alcohols, in combination with endogenous peptides, regulate protein functions via modulating the lipid polymorphism of membranes.

## 1. Introduction

The role of alcohols of different alkyl chain lengths in cellular metabolism and their toxic effects on organelles, cells and body organs has been the subject of studies for decades. Alcohols take part in the esterification pathways. The molecular details of the esterification of hydroxycholesterol and oxysterols and implications related to the advancement of a broad range of pathologies have attracted significant attention in recent years [1,2]. Esterification is a reversible reaction controlled by body enzymes and an accumulation of alcohols along with free fatty acids from lipolysis—a reverse reaction of esterification—in response to an increased energy demand in cells [3], and it may lead to diseases triggered from an increased concentration in alcohols. The repetitive administration of amiodarone, a widely used antiarrhythmic drug, increases lipolysis in adipose tissue leading to liver damage that is attributed to an accumulation of free fatty acids [4], but an accumulation of alcohols, which is another product of lipolysis, is also a well-known reason leading to liver damage and other pathologies.

The effects of ethanol on the cardiovascular system, central and peripheral nervous system, digestive system, liver, pancreas, and skeletal muscles have been extensively studied [5,6,7], due to ethanol being the most addictive and the most consumed drug in human history [8]. A persistent high-dose consumption of ethanol leads to a broad range of diseases including cancer, with the average yearly mortality rate of 10% of the ethanol inflicted diseases [9,10]. The toxic effects of methanol, a highly toxic alcohol [11], on the body and particularly on the cardiovascular system, has also been widely studied to understand the potential hazard from the ingestion of methanol from eating fruits and vegetables and from the intake of methanol when it is mistakenly consumed instead of ethanol [12,13]. Despite a multitude of research studies on methanol, ethanol and other alcohols, understanding the molecular mechanism(s) of alcohols has proven elusive due to the very simple molecular structure of alcohols, which has many potential molecular targets. This creates many leads in alcohol research studies, which makes it difficult to elucidate the molecular details of alcohols’ actions [14].

Although alcohols bind to the plasma membranes of a great variety of cells, the plasma membranes of cardiomyocytes are probably the most affected by the action of alcohols [15]. The physiological effects of alcohols may result from a direct interaction of alcohols with the proteins of biological membranes or from their bilayer-modifying effects which influence the functions of membrane proteins. A study on the bilayer partitioning of alcohols with alkyl chain lengths of 1–16 carbons showed that the bilayer-partitioning potency of short-chain alcohols increases in lipid vesicles made of phosphatidylcholine with the increase in the number of carbon atoms in the alkyl chains, but the bilayer-partitioning potency tapers off at the higher chain lengths as alcohols with a higher chain length tend to form micelles outside the membrane rather than inserting into the membrane’s lipid phase [16]. This supports the notion that the hydrophobic alkyl chain rather than OH group is the major factor in determining the type of alcohols’ intermolecular forces which suggests that the alcohols that insert into a membrane affect the hydrophobic lipid phase of the cell membranes. Interestingly, an anesthetic effect of alcohols on tadpoles, which is likely associated with alcohols binding to the membrane lipid phase, reaches a maximum at 1-dodeconol, with longer alcohols being inactive [17].

Daphnia are commonly used in assessing the cardiotoxicity of a wide range of pharmaceuticals and toxic compounds, including alcohols, as it is relatively easy to monitor changes in the heart rate of these organisms [18]. In this work we assessed the effects of a very low concentration of alcohols of alkyl chain lengths of 1–10 carbon atoms, on the heart rate of daphnia and on the activity of ATP synthase of mitochondria isolated from daphnia, to explore if a correlation exists between the alcohol-triggered changes in the heart rate and the mitochondrial ATP synthase activity. We also tested the combined effects of alcohols with a low concentration of melittin, a bee venom cationic peptide, which exerted no sizeable effects on these parameters. Melittin has been shown to mimic the membrane-active properties of endogenous membrane-active peptides and cardiotoxins (CTs).

CTs are cobra venom membrane-active cationic proteins that have been successfully used to probe the structure and functions of biological and model membranes [19,20,21]. CTs are known for affecting the heart contractility through an induction of changes in the Na^+^/Ca^2+^ transmembrane exchange and for altering the ATP synthase activity through triggering changes in the lipid phase polymorphism of mitochondrial membranes [19,20,21,22,23]. It has been suggested that CTs have evolved from the body’s native membrane-active proteins which modulate the activities of membrane-embedded proteins in the plasma membrane and the membranes of organelles via an induction of changes in the morphology and dynamics of the membrane lipid phase [24]. It has recently been shown that melittin mimics the membrane-active properties of CTs in affecting the structure of the surface interface of neutral and acidic membranes [25,26] and in modulating the ATP synthase activity via a triggering of bilayer to non-bilayer transitions in the lipid phase of inner mitochondrial membranes (IMMs) [27].

In this work, we used melittin for probing the ability of alcohols with alkyl chain lengths of 1–10 carbons to alter the packing and dynamics of phospholipids, as well as to facilitate the melittin-induced polymorphic phase transitions of lipids in model IMM systems of oriented lipid films and unilamellar liposomes, and to test the partial permeabilization of the liposomal membranes. To this end, we employed spin-probe EPR and ^1^H-NMR spectroscopy techniques. Overall, our study shows that the synergistic effects of alcohols on melittin’s actions for modifying the lipid phase polymorphism of model IMM lipid systems aligns well with the combined effects of alcohols and melittin on the heart rate and ATP production of daphnia. The results of this study suggest, for the first time, that alcohols with alkyl chains of 2–8 carbons affect, on the one hand, the membrane surface interface to make it more conducive for melittin-triggered changes in lipid phase polymorphism in the IMM-mimicking model membrane and, on the other hand, facilitate the ATP synthase activity in daphnia, which is required for their higher heartbeat rates in the presence of alcohols and melittin. It is proposed that one of the biological roles of alcohols in cells is linked to modulating the physiological activities of native endogenous membrane-active proteins through changes in the membrane surface interface.

## 2. Materials and Methods

### 2.1. Counting Daphnia’s Heart Rate

Alcohols with chain lengths of 1–10 carbons of the highest purity and bee venom melittin were obtained from Sigma Chemical Co. (St. Louis, MO, USA). The melittin was purified from the trace PLA_2_ and other organic contaminants by cation exchange HPLC on a SCX 83-C-13-ET1 Hydropore column (Rainin Instrument, Woburn, MA, USA) as previously described [28].

The water fleas identified as *Daphnia pulex* were cultured in synthetic pond (SP) water containing 5 g of KCl, 4 g of MgSO_4_, 2.65 g of CaCl_2_, 0.6 g of K_2_HPO_4_, 5 g of NaNO_3_, and 0.44 g of FeCl_3_, per one liter pH 7.2 at 20 °C. The fleas were fed with a suspension of *Selenastrum* sp. every other day and kept out of direct sunlight. Every week, 1/3 of the water was changed without disturbing the eggs at the bottom of the tank and the water was aerated daily. For the heartbeat count, individual fleas were placed in a drop of SP water on a depression slide coated in petroleum jelly to immobilize the flea. The number of beats per minute (bpm) was counted at 100× *g* magnification 60 s after the flea was immobilized. To examine the effects of alcohols and melittin on the daphnia’s heart rate, the test solutions of SP water containing 1.65 × 10^−3^ M of one of the alcohols with chain lengths of 1–10 carbons, or 1.65 × 10^−4^ M melittin, or mixtures of 1.65 × 10^−3^ M of one of the alcohols and 1.65 × 10^−4^ M melittin, were prepared. After an individual flea in a drop of SP water was immobilized on a petroleum jelly coated depression slide, the SP water was withdrawn and replaced with the test solution and the bpm counted 60 s after the SP water was replaced with the test solution. Each data point was derived from at least three bpm counts, with each count from an individual flea. The standard deviation was within 5.0% of the means.

### 2.2. Isolation of Mitochondria and Measuring ATP Production in Mitochondrial Samples

For each batch of mitochondria, 100 g of *Daphnia pulex* cultured in SP water was washed twice, resuspended in a buffer of 30% ^2^H_2_O and 70% ^1^H_2_O (5 g/L KCl, 4 g/L MgSO_4_, 2.65 g/L of CaCl_2_, and 5 g/L of NaNO_3_, with a pH 7.2) and ground with a Teflon pestle homogenizer in a glass sleeve. The cellular debris were pelleted by centrifugation at 200× *g*. The mitochondria were then isolated by the sequential centrifugation steps as we have previously described [29,30]. A final protein concentration in the mitochondrial samples in the SP water was 15.7 mg/mL. The phospholipid concentration in the mitochondrial samples—determined at 20 °C by ^31^P NMR using a Bruker AM-300 spectrometer equipped with a temperature control device (Denver, CO, USA) as previously described [30]—was 1.65 × 10^−2^ M; the variation in concentration between the three ^31^P NMR technical replicates was 6%. Mitochondrial ATP production in the daphnia in the absence and presence of melittin and alcohols with chain lengths of 1–10 carbons was measured at 20 °C using a protocol previously described [31] with minor modifications. The control mitochondrial samples (0.5 mL) were incubated with 0.5 mL of 1.5 mM succinate in SP water for 10 min at 20 °C. Aliquots of the mitochondrial samples (0.5 mL) were incubated with 1.5 mM succinate and 1.65 × 10^−3^ M of alcohols with chain lengths of 1–10 carbons or 1.65 × 10^−4^ M melittin or a mixture of alcohols (1.65 × 10^−3^ M) and melittin (1.65 × 10^−4^ M) in 0.5 mL of SP water for 10 min at 20 °C. To stop ATP synthesis, the mitochondria were lysed in 5 mL of 0.8% Triton X-100, 1 mM D-luciferin, 20 μg *Photinus pyralis* luciferase, 10 mM MOPS buffer, 5 mM MgSO_4_, 0.1 M ethylenediaminetetraacetic acid, and 1 mM dithiothreitol at pH 7.2 by a 10 min incubation at 20 °C. The luminescence intensities were recorded at 20 °C using a luminometer equipped with a temperature control device (Berthold Technologies, Oak Ridge, TN, USA). The luminescence intensities were then converted to the amounts of ATP based on a curve of luminescence intensities from the standard amounts of ATP. To confirm that the ATP in the mitochondrial samples was produced by mitochondrial ATP synthase, an aliquot of mitochondria from the same stock was treated with 3.5 mM oligomycin, an irreversible inhibitor of ATP synthase, which inhibited 90% of the ATP production compared with the control samples. The data points represented mean values from at least three independent experiments. The standard deviation was within 4.5% of the means.

### 2.3. 5-DSA EPR Measurements of Lipid Packing and Fluidity in Oriented Lipid Membranes

The effects of alcohols with chain lengths of 1–10 carbons and of melittin on the structure and dynamics of the model IMM—with a phospholipid composition resembling that of IMM in eukaryotes, i.e., 40 mol% phosphatidylcholine (PC), 35 mol% phosphatidyl-ethanolamine (PE), 20 mol% cardiolipin (CL), 3 mol% phosphatidylinositol (PI) and 2 mol% phosphatidylserine (PS) [32]—was investigated by EPR spectroscopy of spin labeled probe 5-doxylstrearic acid (5-DSA) in oriented model IMM films. The egg yolk L-α-PC, 3-sn-PE from bovine brain, CL from *E. coli*, L-α-PI from soybean, bovine brain L-α-PS, and 5-DSA were purchased from Sigma Chemical Co. (St. Louis, MO, USA). The phospholipids were purified from their residual contaminants on silica columns. Using the above phospholipid composition to prepare the oriented model IMM films by the common technique of squeezing large unilamellar liposomes between two glass plates [33,34], did not yield high quality bilayer films; they did not generate highly resolved EPR spectra of 5-DSA featuring a sharp angular anisotropy [33]. We, therefore, developed a novel technique to prepare the oriented films of the model IMM. In brief, we applied 20 µL of 5.5 × 10^−3^ M phospholipids and 5.5 × 10^−5^ M 5-DSA in 98% ethanol onto a high surface-quality glass plate and allowed it to dry in a helium atmosphere for 60 min at room temperature. We then hydrated the dry lipid film for 60 min in the helium atmosphere with s 20 µL buffer (2 mM Tris-HCl, pH 7.5, and 0.1 mM EDTA) containing either 5.5 × 10^−4^ M alcohol, 5.5 × 10^−5^ M melittin or a mixture of alcohol and melittin, and then covered the hydrated lipid film on a glass plate with another high surface-quality glass plate. The orientation of the hydrated model IMM lipid films in the applied magnetic field was controlled with the resonator accessory. A Varian E-4 spectrometer equipped with a temperature control device (Varian Inc., Palo Alto, CA, USA) was used to record the EPR spectra of the 5-DSA in the oriented IMM films. The EPR spectra were recorded at modulation amplitudes not exceeding 2 × 10^−4^ T and with a resonator input power not exceeding 20 mW at 20 °C. The EPR spectra were analyzed using the parameter *S* [35] and the *B*/*C* ratio [36]. The parameter *S* was calculated according to the formula previously described [35]. *B* is defined as the intensity of the low-field component, while *C* is defined as the intensity of the central component of the EPR spectra of oriented IMM films taken with respect to the applied magnetic field perpendicular to the normal of the IMM film. Each sample for the EPR assay was prepared and tested in triplicate, thus, each data point was the mean derived from the three independent experiments and calculations of the *S* parameter and *B*/*C* ratio. The standard deviation was within 3.5% of the means.

### 2.4. The ^1^H NMR Study on Changes in Membrane Permeability and Structure of Sonicated Unilamellar Liposomes

To investigate the effects of melittin and alcohols with chain lengths of 1–10 carbons on the permeability and morphology of the model IMM lipid membranes—sonicated unilamellar liposomes—^1^H NMR spectroscopy was employed in the presence of potassium ferricyanide K_3_[Fe(CN)_6_]. The sonicated unilamellar liposomes were made of 40 mol% PC, 35 mol% PE, 20 mol% CL, 3 mol% PI and 2 mol% PS to mimic the phospholipid composition of IMM in eukaryotes [32]. The high surface curvature of the small unilamellar liposomes resembled the surface curvature of cristae tips and inter-cristae membrane contacts [37]. The liposomes were prepared according to a protocol previously published [38] with minor modifications. Phospholipids in chloroform were dried in a vacuum for 1.5 h at room temperature to form a lipid film, which was hydrated in a ^2^H_2_O buffer of 10 mM Tris-HCl, pH 7.4, and 0.5 mM EDTA. The lipid suspension was sonicated with an ultrasonic disperser, USDN-1, Biomed. Engineering Tech. (St. Petersburg, Russia), at a frequency of 22 kHz for 15 min in a helium atmosphere at 4 °C. To remove the heavy phospholipid aggregates, the liposomes were centrifuged at 200× *g* for 60 min and then incubated in the helium atmosphere for 15 h at 10 °C. The phospholipid concentration in the liposomes was 1.20 × 10^−2^ M. The liposomes were incubated at 20 °C for 30 min prior to adding 10 µL of a saturated solution of K_3_[Fe(CN)_6_] in ^2^H_2_O to 1 mL of liposomes. Prior to recording the ^1^H-NMR spectra, the liposomes were treated with either 1.20 × 10^−3^ M alcohols, 1.20 × 10^−4^ M or 1.20 × 10^−3^ M melittin or a mixture of alcohol and melittin (the alcohols and melittin were diluted in ^2^H_2_O). The ^1^H-NMR signals from the N^+^(CH_3_)_3_ groups of PC in the liposomes were recorded at 20 °C with an operating frequency of 200 MHz with the Varian XL-200 spectrometer equipped with a temperature control device (Varian Inc., Palo Alto, CA, USA). The width of a 90° pulse was 8.7 µs and the acquisition time for the free induction signal was 1 s. The membrane permeability of the liposomes was assessed by measuring the ^1^H-NMR signal areas from the N^+^(CH_3_)_3_ groups of PC in the outer (I_o_) and the inner (I_i_) leaflets of the liposomal membrane as previously described [28,39,40]. The molar percentage of non-bilayer-organized PC molecules from the total number of moles of PC in the liposomes treated with melittin and alcohols was assessed by calculating the percentage of the computer-extrapolated area under the non-bilayer ^1^H-NMR signal from the overall area of the ^1^H-NMR spectrum, including signals from the I_o_, I_i_ and non-bilayer-organized PC molecules using a Gaussian fitting as previously described [29,36,41,42]. The experiments and measurements for each data point were completed in triplicate; hence, the data points represent the means from three experiments and measurements. The standard deviation for the membrane permeability and percentage of non-bilayer-organized phospholipids data was within 5.0% of the means.

### 2.5. Statistics

Data are presented as mean ± standard deviation (SD). A Student’s *t*-test was used for analysis of the continuous variables. In addition, an Anova test was completed for statistical analysis of the two sets of trials completed by different experimental methods: in one set an “alcohols only” against “alcohols + melittin” trial (Appendix A) and in the other set, a melittin concentration of 1.2 × 10^−4^ M against a melittin concentration of 1.2 × 10^−3^ M trial (Appendix A). A *p*–value < 0.05 was considered as a statistically significant difference.

## 3. Results

### 3.1. Isolated and Combined Effects of Alcohols and Melittin on the Heart Rate of Daphnia pulex

To study the effects of alcohols on the heart rate of daphnia, we used alcohols at a 1.65 × 10^−3^ M concentration, at which methanol, the smallest alcohol, triggered the minimum change in the heart rate of the daphnia. With an increase in the number of carbon atoms from methanol to octanol, the alcohols caused a decrease in the heart rate (Figure 1); however, starting from the nonanol, the effects of alcohols on the heart rate became less pronounced, namely, the heart rate values started increasing to the level of untreated daphnia (Figure 1).

Melittin was used at 1.65 × 10^−4^ M, a concentration that caused merely a very small increase in the heart rate of daphnia and the same was true in the presence of methanol (Figure 1); however, other alcohols, from ethanol to octanol, gradually increased (with an increase in the number of carbon atoms), the heart rate of the daphnia treated with the melittin. At the same time, nonanol and decanol, which affected the daphnia’s heart rate to a lesser extent, enabled a melittin-triggered increase in the daphnia’s heart rate to a lesser extent than the other alcohols with chain lengths from 2–8. It appears that the alcohols that impeded the heart rate of the daphnia also enabled the melittin-triggered increase in the heart rate.

### 3.2. Isolated and Combined Effects of Alcohols and Melittin on Daphnia pulex ATP Synthesis

As above, for the heart rate, we used the same concentrations of alcohols and melittin, 1.65 × 10^−3^ M and 1.65 × 10^−4^ M, respectively, to study their isolated and combined effects on the ATP production in mitochondria isolated from *Daphnia pulex*. Similar to the effects of alcohols on the heart rate, alcohols from ethanol to octanol gradually decreased the ATP production of *Daphnia pulex* mitochondria, but with the nonanol and decanol the ATP production started to increase and came close to the level of ATP produced in the untreated mitochondria of daphnia and of the daphnia treated with only methanol (Figure 2). Melittin alone insignificantly increased the ATP production of mitochondria. When the mitochondria were treated jointly with melittin and methanol, the ATP production was slightly higher than that in the presence of melittin only. Nonanol and decanol also enabled the melittin-triggered increase in ATP production but to a lesser extent than the other alcohols (from ethanol to octanol). Interestingly, the modes of action of the alcohols and melittin, isolated and combined, on the heart rate and the ATP production were very similar. Apart from the methanol, the alcohols inhibited the heart rate and the ATP production, but in combination with melittin they increased both the heart rate and the production of ATP. It should be noted that the stronger the alcohols inhibited the heart rate and the production of ATP, the stronger they facilitated the melittin-induced increases in the heart rate and the ATP production.

### 3.3. Isolated and Combined Effects of Alcohols and Melittin on the Lipid Packing and Dynamics of Model IMM

To examine the effects of alcohols and melittin on the packing order and dynamics of lipids in lipid-membrane models with phospholipid composition closely resembling that of IMM [28], we used the method of an EPR of 5-DSA spin probe in oriented lipid films. The 5-DSA was a stearic acid with the doxyl spin probe attached to the fifth carbon atom of the alkyl chain. In the bilayer membrane of liposomes, which are squeezed into lipid bilayer films between two glass plates, the long molecular axis of 5-DSA tends to align along the normal to the surface [33]. This generates strong molecular anisotropic dynamics of 5-DSA that produces highly resolved EPR spectra with different widths of the spectral lines when the molecular axis of 5-DSA is oriented parallel or perpendicular to the direction of the magnetic field. However, as pointed out above, the EPR spectra of the 5-DSA in the oriented films prepared by squeezing the liposomes of the phospholipid composition of the IMM did not generate a strong enough angular anisotropy of the EPR spectra. Thus, we applied the novel procedure of preparing the oriented hydrated film between two high surface-quality glass plates (see Section 2.3), which produced a well discernible angular spectral anisotropy of the 5-DSA. The spectral line of the 5-DSA recorded in the applied magnetic field parallel to the film’s normal was wide, while a spectral line of the 5-DSA in the applied magnetic field perpendicular to it was narrow (Figure 3A). This, along with the long axis of the 5-DSA aligned along the normal to the film, indicates a bilayer packing of phospholipids in the hydrated film [27].

Treatment of the lipid film with melittin reduced, to some degree, the angular spectral anisotropy of the 5-DSA, as shown by the broadening of the EPR spectral line taken at the parallel orientation of the applied magnetic field, and the appearance of a broad low-field component in the EPR spectra taken at the perpendicular orientation of the applied magnetic field (Figure 3B). Treatment of the lipid film with methanol, nonanol and decanol in the absence of melittin did not cause visible changes in the spectral lines of the 5-DSA at parallel and perpendicular orientations of the lipid films in the applied magnetic field (spectra not shown). A gradual increase in the disturbance of the angular spectral anisotropy of the 5-DSA was observed when the lipid films were separately treated with alcohols with an increasing number of carbon atoms in their alkyl chains from ethanol to octanol (spectra not shown). When the effect of ethanol on the spectral anisotropy of the 5-DSA was barely visible, the spectral lines obtained at the parallel and perpendicular orientation of the long axis of the 5-DSA in the lipid films treated with octanol, closely resembled the spectral lines in Figure 3B.

The combined treatment of the lipid film with melittin and ethanol further reduced the spectral anisotropy of the 5-DSA, as revealed by the prominent low-field and high-field components of the EPR spectrum taken at the perpendicular orientation of the applied magnetic field; the resonance frequencies of which closely coincided with the same components in the EPR spectrum taken at the parallel orientation of the applied magnetic field (Figure 3C). Similar effects on the EPR spectral lines of the 5-DSA in the lipid films treated jointly with alcohol and melittin were observed with propanol to octanol, while the effects of methanol, nonanol and decanol were virtually non-apparent (spectra not shown).

To quantitatively assess the effects of alcohols and melittin on the lipid packing and polymorphism in the lipid films, we calculated the *B*/*C* ratio from the 5-DSA EPR spectra. The *B*/*C* ratio is sensitive to the macroscopic lipid packing in membranes [33,39,41]. Values around 0.7 to 0.8 indicate a highly ordered bilayer packing of lipids [33,41] while *B*/*C* ratios below 0.4 indicate the formation of non-bilayer lipid structures [33,39,41]. As one can see from Figure 4, treatment of the lipid film with alcohols, from ethanol to octanol, decreased the *B*/*C* ratio from 0.56 to 0.45, which most likely reflects local disturbances from the membrane surface interface towards the fifth carbon atom of the alkyl chains; however, this does not cause a transition from bilayer to non-bilayer packing of lipids [33,41]. The effects of methanol on the *B*/*C* ratio were negligible and the same of nonanol and decanol were significantly less pronounced than that of the alcohols from ethanol to octanol (Figure 4).

Melittin in the absence of alcohols caused a decrease of the *B*/*C* ratio from 0.65 to 0.46 (Figure 4), which suggests that it disturbed the surface area of the membrane deep to the fifth carbon atom of the alkyl chains without inducing non-bilayer structures; however, alcohols from ethanol to octanol caused a synergistic effect on the melittin-induced formation of non-bilayer structures, as reflected by the decrease of the *B*/*C* ratio values below 0.4, gradually decreasing from 0.37 to 0.26 (Figure 4). Methanol, nonanol and decanol did not synergize with the melittin in the formation of non-bilayer structures.

Parameter *S* was another parameter, which we calculated from the EPR spectra of the 5-DSA in the lipid films with a phospholipid composition of the IMM. *S* is sensitive to changes in lipid dynamics, particularly in the rotational movement of 5-DSA [33,39,42]. An increase in *S* indicates a decreased rotational movement in 5-DSA. At a neutral pH, 5-DSA models the dynamics of acidic phospholipids and free fatty acids in the membrane. As one can see in Figure 5, methanol caused virtually no effect and nonanol and decanol had small effects on the parameter *S*. Other alcohols, from ethanol to octanol, caused a noticeable gradual decrease in the parameter *S*, which points to an increase in rotational movement of the 5-DSA that in turn suggests an overall loosening in the packing of both acidic CL, PI and PS, and zwitterionic PC and PE phospholipids in our model IMM.

Melittin caused an increase in *S*, which is attributed to an electrostatic interaction between the basic melittin and the acidic 5-DSA, decreasing the rotational movement of the 5-DSA. Obviously, the melittin also interacted electrostatically with the acidic phospholipids in the lipid film and decreased the rotational movement of the acidic phospholipids. The methanol had no effect on the interaction of melittin with the membranes (Figure 5); however, other alcohols, from ethanol to octanol, increased the *S* values in the lipid films treated with melittin, which was probably due to local disturbances in the membrane surface interface caused by the alcohols, which helped cationic melittin to overcome electrostatic repulsion from the -N^+^(CH_3_)_3_ groups of PC, the most abundant phospholipid in the IMM, and to reach out to the -PO_4_^−^ groups of PC. It should be noted that nonanol and decanol insignificantly increased the *S* value, compared to the control and the methanol, in the lipid films treated with melittin (Figure 5).

### 3.4. Isolated and Combined Effects of Alcohols and Melittin on the Permeability and Lipid Polymorphism in Unilamellar Liposomes

The ^1^H-NMR spectroscopy of unilamellar liposomes, obtained by sonication, had previously been extensively used by us to study the effects of cobra venom CTs on the permeability and on the bilayer to non-bilayer transitions of phospholipids in liposomal membranes [28,36,38,39,40,42]. Ferricyanide ions, Fe(CN)_6_^3^^−^, are paramagnetic ions and when added to the PC containing liposomes they only interacted with the ^1^H atoms of N^+^(CH_3_)_3_ groups of the PC on the outer leaflet of liposomes, which shifted the ^1^H-NMR signal of the N^+^(CH_3_)_3_ groups to a higher field. At the same time, the ^1^H-NMR signal of the N^+^(CH_3_)_3_ groups of PC in the inner leaflet of liposomes was not shifted, as the Fe(CN)_6_^3^^−^ ions did not penetrate through the membrane of intact liposomes. This resulted in splitting the ^1^H-NMR signals into two peaks, from the inner (I_i_) and outer (I_o_) leaflets of liposomes, respectively (Figure 6A,B).

The addition of cobra venom CTs to PC liposomes that are enriched with acidic phospholipids, but not CL, degrades the structural integrity of the liposomal membrane, rendering the membranes permeable to Fe(CN)_6_^3^^−^ ions, which can then gain access to the N^+^(CH_3_)_3_ groups in the inner leaflet of a liposomal membrane. This shifts the ^1^H-NMR signal also from the inner leaflet to a higher field, which makes the ^1^H-NMR signals from the outer and inner leaflets of liposomes indistinguishable, as they then resonate in the same field. We have observed previously that the addition of low concentrations of cobra CTs to PC liposomes containing 2.5, 5 or 10 mol% CL [39,42] has resulted in the formation of a new ^1^H-NMR signal on the high-field shoulder of the signal from the outer leaflet (I_o_ signal), similar to that shown in Figure 6C–E. We have shown that this new signal originates from the PC in non-bilayer packed phospholipid clusters that apart from PC also contain CL; the formation of which is triggered by cobra CT [39,42]. The packing of PC in non-bilayer clusters, which have a higher concentration of Fe(CN)_6_^3^^−^ ions, permits a closer interaction of the N^+^(CH_3_)_3_ groups with Fe(CN)_6_^3^^−^ ions than that in bilayer structures; this shifts the ^1^H-NMR signal from the PC in non-bilayer clusters further to the higher field [39,42].

In this study we used Fe(CN)_6_^3^^−^ ions-treated unilamellar liposomes with a phospholipid composition mimicking that of eukaryotic IMM to study the effects of melittin and alcohols, not only on membrane permeability, but also on whether PC molecules are involved in the non-bilayer transitions in the model membrane of IMM. The ^1^H-NMR spectrum of liposomes in the presence of Fe(CN)_6_^3^^−^ ions’ sample prepared in this study, displayed two well discernible signals from the inner (I_i_) and the outer (I_o_) leaflets, respectively, of the liposomal membrane (Figure 6A). When the liposomes were separately treated with methanol, ethanol, propanol, butanol, pentanol, hexanol, heptanol, octanol, nonanol or decanol at a lipid to alcohol molar ratio of 10 to 1, the ^1^H-NMR spectra of all liposome samples in the presence of Fe(CN)_6_^3^^−^ ions closely resembled the ^1^H-NMR spectrum in Figure 6A. The addition of melittin to liposomes at a lipid to melittin molar ratio of 100 to 1 resulted in a slight broadening of the ^1^H-NMR signals, which was likely caused by a melittin-triggered mild aggregation of liposomes [40]; however, the structural integrity of the liposomal membrane was not affected and the membrane remained impermeable to Fe(CN)_6_^3^^−^ ions (Figure 6B). When the liposomes contained melittin (with a lipid to melittin molar ratio of 100 to 1) and were separately treated with either methanol, nonanol or decanol (with a lipid to alcohol molar ratio of 10 to 1), the ^1^H-NMR spectra also closely resembled the ^1^H-NMR spectrum in Figure 6B. The addition of both melittin (with a lipid to melittin molar ratio of 100 to 1) and ethanol to the liposomes (with a lipid to ethanol molar ratio of 10 to 1), resulted in the formation of a small non-bilayer (nb) signal, but the liposomal membrane remained impermeable to Fe(CN)_6_^3^^−^ ions (Figure 6C). The same was observed in liposomes containing melittin but also treated separately with propanol, butanol, pentanol, hexanol, heptanol (spectra not shown) and octanol (Figure 6D). In all these samples, the liposomal membrane remained impermeable to Fe(CN)_6_^3−^ ions, but the non-bilayer (nb) signal gradually increased from the ethanol (Figure 6C) to the octanol (Figure 6D). When the liposomes were treated with a high concentration of melittin (with a lipid to melittin molar ratio of 10 to 1) and the same concentration of ethanol (with a lipid to ethanol molar ratio of 10 to 1), the ^1^H-NMR spectrum displayed a high intensity non-bilayer (nb) signal and a weak inner leaflet signal (I_i_) (Figure 6E), revealing that the liposomal membrane became permeable to Fe(CN)_6_^3−^ ions. It should be noted that the ^1^H-NMR spectrum of liposomes in the absence of alcohols but treated with a high concentration of melittin (with a lipid to melittin molar ratio of 10 to 1), closely resembled the spectrum in Figure 6E. The same ^1^H-NMR spectra were observed when liposomes containing a high concentration of melittin (with a lipid to melittin molar ratio of 10 to 1) were separately treated with methanol and other alcohols, from propanol to decanol (spectra not shown). This observation suggests that at a high melittin concentration, alcohols do not visibly contribute to the formation of the non-bilayer phase in liposomes with the phospholipid composition of the IMM.

To quantitatively assess the effects of melittin and alcohols on the permeability and the bilayer to non-bilayer transitions in the lipid phase of the liposomal membrane, we calculated the I_i_/I_o_ ratio and the ratio of the inner leaflet signal area (I_i_) to the outer leaflet signal area (I_o_), and assessed the molar percentage of non-bilayer organized PC in the liposomes by calculating the percentage of the computer-extrapolated area under the non-bilayer signal from the area under the entire spectrum that included the I_o_, I_i_ and non-bilayer signals. The results of our calculations are shown in Table 1.

A decrease in the I_i_/I_o_ ratio means an increase in the permeability of the liposomal membrane to Fe(CN)_6_^3^^−^ ions. As shown in Table 1, the addition of melittin to the liposomes at a lipid to melittin molar ratio of 100 to 1, slightly increased the I_i_/I_o_ ratio, which was probably caused by a decrease in the mobility of PC molecules in the outer leaflet of the liposomes, due to the interaction of melittin with the surface of the outer monolayer leading to a larger decrease in the I_o_ signal area than in the I_i_ signal area. The addition of alcohols, from methanol to decanol at a lipid to alcohol molar ratio of 10 to 1, in combination with melittin (with a lipid to melittin molar ratio of 100 to 1), did not change the I_i_/I_o_ ratio—indicating that the liposomal membranes in these systems were also not permeable to Fe(CN)_6_^3^^−^ ions. However, the addition of melittin to the liposomes at a lipid to melittin molar ratio of 10 to 1 decreased the I_i_/I_o_ ratio from 0.524 to 0.182 (Table 1), which indicates that melittin alone at a higher concentration is capable of degrading the structural integrity of the liposomal membrane, making it permeable to Fe(CN)_6_^3^^−^ ions. The combined action of the higher concentration of melittin (with a lipid to melittin molar ratio of 10 to 1) and the alcohols from methanol to decanol added separately to the melittin-treated liposomes at a lipid to alcohol molar ratio of 10 to 1, did not further increase the permeability of the liposomal membrane, as the I_i_/I_o_ ratio remained within the values 0.181 to 0.182 (Table 1). This most likely indicates that, at a high concentration, melittin does not need assistance from alcohols in permeabilizing the model membrane of IMM.

Melittin at a lipid to melittin molar ratio of 100 to 1 did not induce the non-bilayer structures in liposomal membranes (Table 1). Additionally, methanol, nonanol and decanol did not facilitate the formation of non-bilayer structures in liposomes treated with melittin at a lipid to melittin molar ratio of 100 to 1; however, other alcohols (ethanol to octanol) gradually increased, with an increase in the carbon number in the alcohol chains, the molar percentage of non-bilayer organized PC from 14.3% to 25.1% in the liposomes treated with melittin at a lipid to melittin molar ratio of 100 to 1 (Table 1). These results prove that in liposomes which resemble the phospholipid composition of IMM, alcohols with 2–8 carbon atoms facilitate the formation of non-bilayer lipid structures triggered by a small concentration of melittin. However, it should be noted that in the same liposomes, when treated with a high melittin concentration (with a lipid to melittin molar ratio of 10 to 1) in the absence of alcohols, about 37 molar percent of PC molecules were non-bilayer organized. The combined action of melittin and alcohols (from methanol to decanol) did not further increase the amount of non-bilayer organized PC molecules. This observation indicates that, at a high melittin concentration, the contribution of alcohols in facilitating the non-bilayer lipid phase transition is virtually non-existent, a conclusion which is also supported by an Anova test *p*-value nearing 1 (Appendix A), pointing to the ‘statistical’ absence of difference in the percent of non-bilayer organized PC molecules in model IMM liposomes treated with a high concentration of melittin alone or jointly with alcohols.

## 4. Discussion

The biological activity of alcohols in cells, their role in cellular metabolism, their toxic effects on organelles, cells and organs and their potential pharmaceutical activities have been extensively investigated for many decades [5,6,7,8,9,10,14,15,16], yet the molecular mechanism(s) of alcohols remain controversial. Alcohols are very simple molecules and have many different potential molecular targets. This creates many leads in tracking alcohols’ modes of action which pose a special challenge in elucidating the molecular mechanism(s) of alcohols [14].

The activities of alcohols in a capacity of being the modulators of biological membrane structure and function are well reported [42,43,44], while the physiological effects of alcohols have been attributed to their bilayer-modifying properties; yet, the question remains if these properties are driven by the ability of alcohols to interact directly with membrane proteins or by their ability to influence structural changes in the lipid phase of biological membranes [16,45].

Cardiovascular systems are intensely investigated concerning the effects of alcohols, especially ethanol [5,6,7,8,9,10] and it is well known that the health of the heart is directly linked to the health of mitochondrial energetics [34]; however, the effects of alcohols on mitochondrial energetics have not been given due attention. In this study, we performed parallel experiments on the effects of alcohols on the heartbeat and the mitochondrial ATP production of daphnia, a eukaryotic organism which has a heart very similar to the vertebrate myogenic heart. We also studied the combined effects of alcohols with melittin—a bee venom membrane-active polypeptide, which is similar to endogenous membrane-active peptides and is capable of affecting mitochondrial respiration [27]—on the heartbeat and mitochondrial ATP production of daphnia. Further, we investigated the individual and combined effects of alcohols and melittin on the polymorphic lipid phase transitions and the permeability of model membranes of IMM with a phospholipid composition mimicking that of the IMM of eukaryotes.

We found a clear correlation between the effects of alcohols and melittin on the heartbeat rate and the mitochondrial ATP production of daphnia, with gradually increasing effects from ethanol to octanol, and no or considerably smaller effects with nonanol and decanol (Figure 1 and Figure 2). Additionally, in lipid-model systems, methanol, nonanol and decanol caused negligible disturbance in the bilayer packing of lipids, while the bilayer-disturbing effects of other alcohols, from ethanol to octanol, were gradually more significant—although not enough to induce non-bilayer structures (Figure 4, Figure 5 and Figure 6 and Table 1). Melittin alone more severely disturbed the lipid packing and—in synergy with the alcohols, from ethanol to octanol—induced non-bilayer lipid phases (Figure 4 and Figure 6 and Table 1). At the same time, melittin in combination with the same alcohols which gradually decreased the order parameter *S* of the spin-labeled lipid films, gradually increased the *S* values (Figure 5) but without permeabilizing the liposomal membrane (Table 1). These observations might indicate the insertion of melittin into the bilayer, or the formation of a non-bilayer lipid phase outside the bilayer phase, around the water soluble peptide melittin.

We have recently demonstrated that melittin causes the formation of non-bilayer lipid structures which predominantly involve CL molecules in model IMM at a lipid to melittin molar ratio of 67 to 1 [27]. In this study, we demonstrated for the first time that alcohols with alkyl chains containing 2–8 carbon atoms trigger the formation of non-bilayer structures in model IMM at a lipid to melittin molar ratio of 100 to 1. We have also demonstrated for the first time that non-bilayer structures triggered in model IMM by a combined action of melittin (with a lipid to melittin molar ratio of 100 to 1) and alcohols, from ethanol to octanol, (with a lipid to alcohol molar ratio of 10 to 1), involve not only CL [27] but also PC molecules. It is also important to stress that the formation of non-bilayer structures in our melittin-alcohol-model IMM systems did not disrupt the barrier properties.

It is noteworthy that methanol, nonanol and decanol did not significantly facilitate the action of melittin on the daphnia’s heartbeat and mitochondrial ATP production or on the structural changes in the model IMM—as opposed to other short-chain alcohols. Overall, the alcohols’ efficacies investigated in this study gradually increased with an increase in the number of carbon atoms until nonanol, after which an abrupt efficacy decrease was observed. This mode of action of the alcohols agrees well with the previously reported, so-called cutoff effect, in which alcohols exhibit significantly lesser biological efficacy at a certain length of alkyl chain (which varies from 8–14 carbon atoms in an alkyl chain depending on the experimental conditions) [16,46,47]. The cutoff effect appears to occur when the alcohol’s alkyl chain length is about equal to half of the alkyl chain length of the bilayer lipids [16,44,48] and this could be due to the interplay between the alcohol’s affinity binding to a membrane and the energy released when alcohols assemble into micelles, which increases with the increase in length of an alcohol’s alkyl chain [16,48].

The important role of non-bilayer structures in ATP production and in the maintenance of the structural integrity of IMM has been recently reported by us not only for mitochondria [30,34,49], but also in fully functional plant thylakoid membranes [49]. Non-bilayer structures are not only important in membrane fusion and the intermembrane exchange of lipids, which are essential processes in maintaining the functional dynamic architecture of the highly folded IMM [34,49], but they are also key elements in the formation of sub-compartments in the inter-cristae space near the ATP synthases, whereby they play an important role in enhancing the proton motive force and ATP synthase activity [30,34,49].

As concerns the involvement of PC in the melittin-triggered and alcohol-facilitated formations of non-bilayer structures in model IMM, it occurred at a lipid to melittin molar ratio of 100 to 1. It is important to note, however, that in the absence of alcohols, no non-bilayer structures were formed (Figure 5, Table 1). Additionally, in the absence of alcohols, the melittin induced only an insignificant increase in the heartbeat and ATP production of the daphnia. Obviously, to increase the heartbeat rate, more ATP must be produced, which—as inferred from the data in [30]—requires a greater amount of non-bilayer structures in the IMM. This can be achieved via an interaction of melittin with the CL of the IMM, but to reach the IMM, the melittin must penetrate through the cellular plasma membrane and the outer mitochondrial membrane (OMM). The outer leaflet of both the plasma membrane and the OMM are predominantly made of PC and although the PC (which has basic choline, -N^+^(CH_3_)_3_, and acidic phosphate, -PO_4_^−^, groups) is electrically neutral, it is the choline group which is exposed on the membrane surface while the phosphate group lays beneath the choline group, lending the PC membrane surface a predominantly positively charge. Nevertheless, cationic melittin at high concentrations manages to penetrate the PC membrane [25,50].

Melittin, a peptide of 26 residues, has the alpha-helical rod structure that bends at the 14th residue with a 120° angle [51]. The side chains of hydrophobic residues of melittin are oriented to inside of the bend, while the side chains of the charged and polar residues are oriented to the outside of the bend [51]. Melittin approaches the membrane surface with its long molecular axis parallel to the membrane plane with the side chains of charged and polar residues oriented to the membrane plane [25,50,51]. Previously we predicted by an AutoDock simulation that R22 is a key residue that drives melittin binding to a PC membrane via C=N^+^H_2_ and C-NH_2_ ^δ+^ groups of R22, which make the ionic and ion-polar bonds with a phosphate group of the PC. This may occur when the membrane is sufficiently disturbed to expose the phosphate group on the membrane surface [25]. Once the melittin embeds in the lipid polar head area of the PC membrane, it twists along its long molecular axis to orient its hydrophobic residue side chains to lipid alkyl chains and the hydrophilic residue side chains to the lipid polar head area [25,50,51,52]. The localization of melittin on the outer monolayer of the membrane only creates an asymmetrical intermembrane surface tension, which is released at a lipid to melittin molar ratio of 100 to 1 through a reorientation of the melittin’s long molecular axis from parallel to perpendicular to the membrane plane to promote the formation of stable membrane pores [50] and to facilitate the translocation of the melittin across the membrane.

The presence of alcohols disturbs the surface of the PC membrane to create opportunities for an interaction of melittin’s R22 with the phosphate group of the PC, which facilitate melittin translocation through the membrane. Once the melittin reaches the surface of the IMM, it reacts with CL on the membrane surface to trigger a formation of intra-cristae bridges made of non-bilayer-organized CL molecules according to the mechanism described for CTs [22,30,34] and proposed for cyt *c* [49]. In the presence of alcohols, melittin can also react with the phosphate group of PC, which explains the involvement of PC in non-bilayer structure formation, as revealed here by ^1^H-NMR spectroscopy. Involvement of PC into the non-cylinder-shaped CL non-bilayer structures probably decreases the inner-surface curvature of the non-bilayer structures that stabilizes the structures. Thus, we propose that short-chain alcohols, excluding methanol, facilitate melittin’s membrane-active propensity by making a membrane surface more conducive for the melittin-triggered formation of non-bilayer structures.

Cobra venom cationic protein toxins, CTs, which phenocopy the membrane-active capacities of melittin, have been proposed to evolve from endogenous cationic proteins; they regulate the functions of membrane proteins through triggering the fluidity and the polymorphic lipid phase behavior of membranes [24,53]. Lipocortins, a group of glucocorticoid-controlled proteins, regulate phospholipase A_2_ (PLA_2_) activity in vivo [54]. A small cationic peptide derived from the lipocortin family modulates PLA_2_ activity in vitro similarly to cobra venom CTs [24,35,55] and, in parallel, exhibits anti-inflammatory activity in vivo [55]. Another endogenous cationic peptide, isolated from different sources (human arthritic synovial fluid, murine smooth muscle and bovine endothelial cells), regulates PLA_2_ activity [56]—presumably through the modulation of the membrane–lipid substrate interface [35,57]. This endogenous cationic peptide is structurally similar to bee venom melittin and is recognized by anti-melittin antibodies [56]. Another membrane-active peptide which regulates PLA_2_ activity via modulation of the packing of the lipid phase of membranes was isolated from murine embryonic genital tracts [35,57,58]. Two decades ago, a mitochondria-derived peptide, humanin (HN), was identified in the brain of Alzheimer patients [59]. HN, a 24 amino-acid residues peptide [60], is similar in size and structure to melittin. HN counteracts the mitochondrial dysfunction in cardiac tissue by stabilizing the mitochondrial membrane potential, mitochondrial structure and ATP levels [61]—presumably through affecting the mitochondrial membrane lipid phase via an unknown mechanism [62,63,64]. Thus, it is possible to suggest that bee venom melittin evolved similarly to cobra venom CTs from endogenous cationic peptides which regulate the activities of membrane-associated proteins via the modulation of the lipid phase polymorphism of membranes. In this study we have shown that short-chain alcohols facilitate melittin, at a low concentration, to increase the heart rate and the ATP production of daphnia, most likely via the formation of non-bilayer structures in the IMM. This leads us to propose that one of the biological roles of short-chain alcohols in cells is the regulation of activities of the membrane-associated endogenous proteins through inducing changes in the lipid polymorphism of membranes.

It should also be noted that methanol, which could be called a no-chain alcohol, showed virtually no effects on the heartbeat rate and the ATP synthase activity of daphnia and equally showed virtually no effects on the membrane activity of melittin. This suggests that methanol has little affinity to the lipid phase of a membrane, probably due to the lack of a hydrophobic moiety; hence, the well-known strong toxicity of methanol cannot be due to its interaction with the membrane but can very likely be explained by the accumulation of methanol’s toxic metabolites, such as formaldehyde and formic acid [65].

## 5. Conclusions

In this study, we have shown that melittin at a low concentration does not affect the heartbeat rate or the ATP production in daphnia and does not induce non-bilayer structures in model IMM. In contrast, in the presence of short-chain alcohols, it significantly increases both the heart rate and the ATP production of daphnia and induces non-bilayer lipid phase(s) in model IMM. Our data show that short-chain alcohols disturb the ordered packing of the lipid bilayer, making the membrane surface more conducive for melittin infraction with the membrane and the formation of non-bilayer structures. We propose that short-chain alcohols, in combination with endogenous peptides—such as mitochondria-derived humanin, lipocortins-derived anti-inflammatory cationic peptide, human arthritic synovial fluid-derived cationic peptide, murine embryonic genital tracts-derived cationic peptide and cyt *c*—regulate the activities of membrane-associated proteins through the modulation of the lipid phase polymorphism of membranes.

## Figures and Tables

**Figure 1 cells-11-01928-f001:**
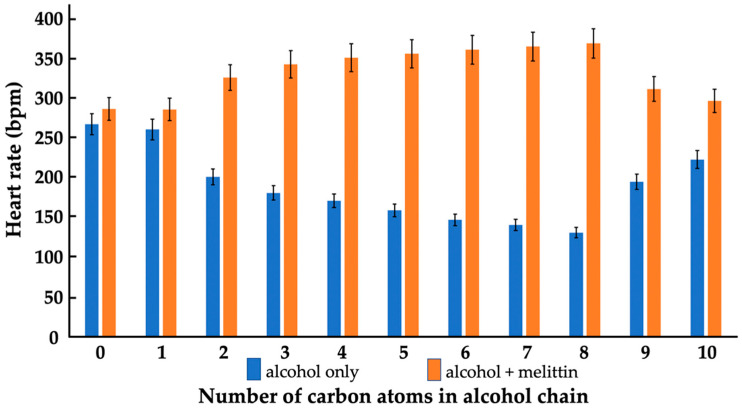
Heart rate of *Daphnia pulex* as a function of alcohol chain length in the absence of melittin and in melittin-treated organisms. The heart rate was measured in beats per minute (bpm). Individual fleas were treated with either alcohols, melittin or a mixture of alcohol and melittin as described in the Materials and Methods. Concentrations of alcohols and melittin were 1.65 × 10^−3^ M and 1.65 × 10^−4^ M, respectively. Blue and orange bars at 0 on the horizontal axis show, respectively, the heart rates of the control sample and sample treated with melittin in the absence of alcohol. Data points represent mean values from at least three independent experiments on different fleas. The standard deviations shown in error bars were within 5.0% of the means.

**Figure 2 cells-11-01928-f002:**
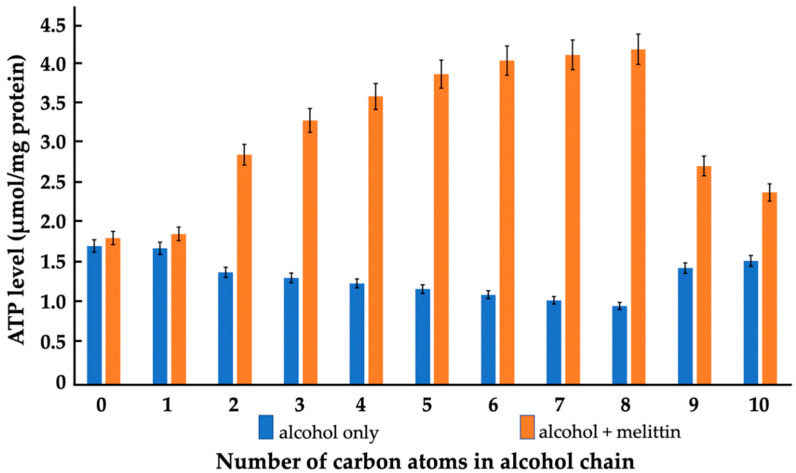
Mitochondrial ATP synthesis of *Daphnia pulex* as a function of alcohol chain length in the absence of melittin and in melittin-treated organelles. ATP levels were assessed as µmol ATP synthesized per mg of mitochondrial proteins. Mitochondria were treated with either alcohols, melittin or a mixture of alcohol and melittin as described in the Materials and Methods. The bulk phospholipid concentration in the mitochondrial samples, estimated via comparing the integrated areas of the ^31^P NMR signals to that of large multilamellar liposomes, was approximately 1.65 × 10^−2^ M. Concentrations of alcohols and melittin were 1.65 × 10^−3^ M and 1.65 × 10^−4^ M, respectively. Blue and orange bars at 0 on the horizontal axis show, respectively, the ATP levels in the control sample and in sample treated with melittin in the absence of alcohol. Each data point is the mean of three independent experiments. Standard deviations, shown as error bars, were within 5.2% of the means.

**Figure 3 cells-11-01928-f003:**
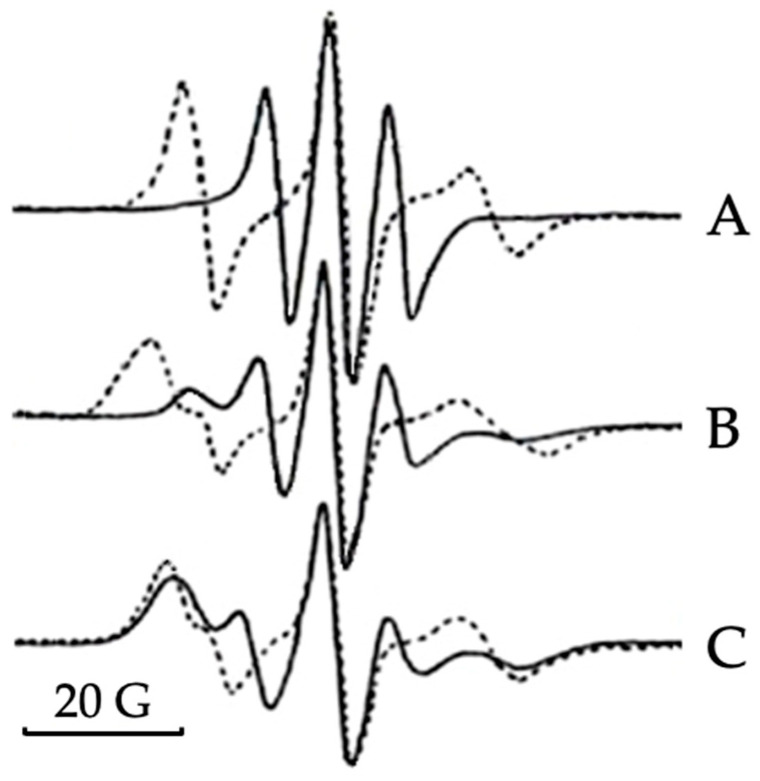
EPR spectra of 5-doxylstearic acid (5.5 × 10^−5^ M) in oriented lipid films—of 5.5 × 10^−3^ M phospholipids concentration, with a composition of 40 mol% PC, 35 mol% PE, 20 mol% CL, 3 mol% PI and 2 mol% PS—at the applied magnetic field parallel (broken line) and perpendicular (solid line) to the film’s normal in the absence of alcohols and melittin (**A**) and in the presence of 5.5 × 10^−5^ M melittin (**B**) and 5.5 × 10^−5^ M melittin and 5.5 × 10^−4^ M ethanol (**C**).

**Figure 4 cells-11-01928-f004:**
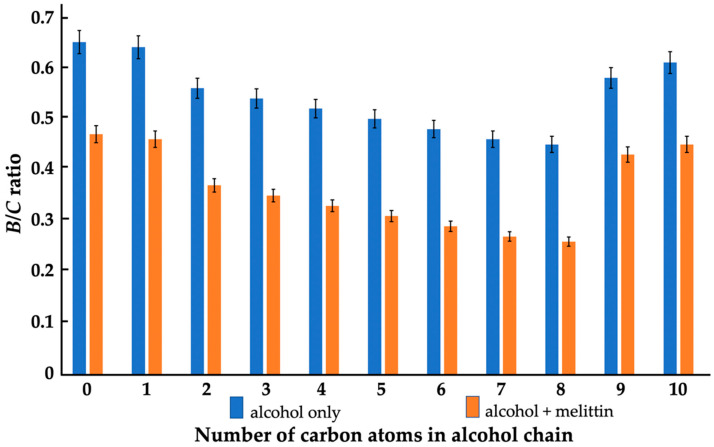
Effects of alcohols (5.5 × 10^−4^ M) in the absence and presence of melittin (5.5 × 10^−5^ M) on the *B*/*C* ratio of the EPR spectra of 5-DSA in oriented lipid films made of a phospholipid composition which mimics that of the IMM in eukaryotes. Concentrations of the total phospholipids and 5-DSA in the lipid films were 5.5 × 10^−3^ M and 5.5 × 10^−5^ M, respectively. Standard deviations, shown as error bars, were within 5.2% of the means.

**Figure 5 cells-11-01928-f005:**
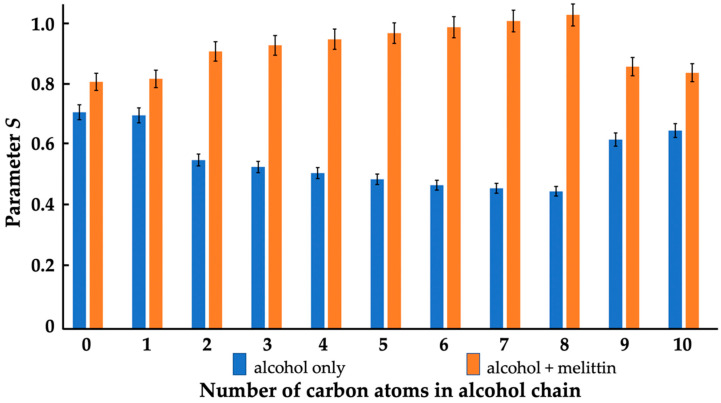
Effects of alcohols (5.5 × 10^−4^ M) in the absence and presence of melittin (5.5 × 10^−5^ M) on parameter *S* of EPR spectra of 5-DSA in oriented lipid films of phospholipid composition, mimicking that of the IMM in eukaryotes. The concentrations of total phospholipids and 5-DSA in lipid films were 5.5 × 10^−3^ M and 5.5 × 10^−5^ M, respectively. Standard deviations shown by error bars were within 5.5% of the means.

**Figure 6 cells-11-01928-f006:**
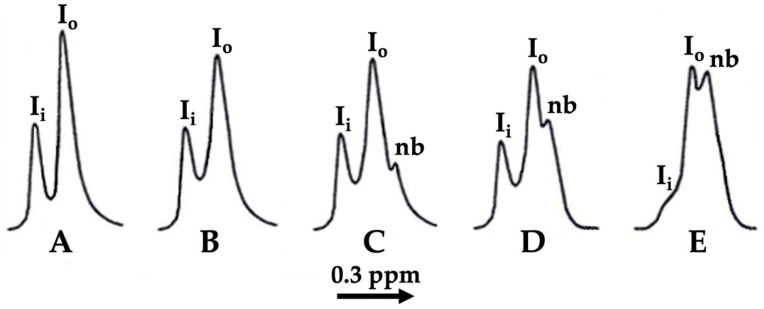
The ^1^H-NMR spectra derived from the N^+^(CH_3_)_3_ groups of PC in the inner (I_i_) and the outer (I_o_) leaflets of unilamellar liposomes, mimicking the lipid composition of eukaryotic IMM, in the presence of K_3_[Fe(CN)_3_]. (**A**) untreated liposomes; (**B**–**E**), respectively, liposomes treated with 1.2 × 10^−4^ M melittin, 1.2 × 10^−4^ M melittin and 1.2 × 10^–3^ M ethanol, 1.2 × 10^−4^ M melittin and 1.2 × 10^−3^ M octanol, and 1.2 × 10^−3^ M melittin and 1.2 × 10^−3^ M ethanol. Total phospholipid concentration, 1.20 × 10^−2^ M.

**Table 1 cells-11-01928-t001:** Increase in membrane permeability to Fe(CN)_6_^3^^−^ ions and the formation of non-bilayer organized PC in the membranes of sonicated unilamellar liposomes treated with melittin and alcohols. The phospholipid composition of liposomes closely resembled that of the IMM in eukaryotes. The permeability was assessed by calculating the I_i_/I_o_ ratio, where I_i_ and I_o_ are, respectively, the inner and the outer leaflet ^1^H-NMR PC signal areas. The molar percentage of non-bilayer organized PC molecules was assessed by calculating the percentage of the computer-extrapolated area under the ^1^H-NMR non-bilayer signal of PC from the overall area of the ^1^H-NMR spectrum, including the signals from the I_o_, I_i_ and non-bilayer organized PC molecules using a Gaussian fitting as previously described [22,29,36,41]. The total phospholipid concentration was 1.2 × 10^−2^ M. In all experiments, the alcohol concentration was 1.2 × 10^−3^ M, while the melittin concentration was 1.2 × 10^−4^ M or 1.2 × 10^−3^ M in the different series of experiments. Each data point was the mean of three independent experiments. Standard deviations for all data points were within 4.7% of the means. Abbreviations: M: melittin, met: methanol, eth: ethanol, pro: propanol, but: butanol, pent: pentanol, hex: hexanol, hep: heptanol, oct: octanol, non: nonanol, dec: decanol, NB (%): molar percentage of non-bilayer-organized PC.

**Melittin Concentration 1.2 × 10^−4^ M**												
	**Control**	**M**	**M + met**	**M + eth**	**M + pro**	**M + but**	**M + pent**	**M + hex**	**M + hep**	**M + oct**	**M + non**	**M + dec**
I_i_/I_o_	0.524 ± 0.012	0.577 ± 0.008	0.576 ± 0.006	0.578 ± 0.013	0.575 ± 0.009	0.578 ± 0.009	0.576 ± 0.004	0.575 ± 0.007	0.574 ± 0.005	0.569 ± 0.006	0.572 ± 0.012	0.571 ± 0.009
NB (%)	—	—	—	14.3 ± 0.35	16.1 ± 0.36	18.1 ± 0.40	19.9 ± 0.36	21.6 ± 0.17	23.4 ± 0.46	25.1 ± 0.26	—	—
**Melittin Concentration 1.2 × 10** **^−3^ M**												
	**Control**	**M**	**M + met**	**M + eth**	**M + pro**	**M + but**	**M + pent**	**M + hex**	**M + hep**	**M + oct**	**M + non**	**M + dec**
I_i_/I_o_	0.524 ± 0.012	0.182 ± 0.003	0.182 ± 0.003	0.181 ± 0.003	0.181 ± 0.004	0.181 ± 0.003	0.181 ± 0.004	0.181 ± 0.003	0.181 ± 0.003	0.181 ± 0.001	0.182 ± 0.003	0.182 ± 0.002
NB (%)	—	36.9 ± 0.85	36.9 ± 0.70	37.0 ± 0.85	37.0 ± 0.82	37.0 ± 0.72	37.0 ± 0.72	37.0 ± 0.56	37.0 ± 0.87	37.0 ± 0.62	36.9 ± 0.79	36.9 ± 0.75

## Data Availability

All data obtained in this study are contained within the article.

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
