# Peer review of "Short-Chained Alcohols Make Membrane Surfaces Conducive for Melittin Action: Implication for the Physiological Role of Alcohols in Cells"

_cells, 2022, doi:10.3390/cells11121928_

Round 1
Reviewer 1 Report
The work of Edward S. Gasanov's group is devoted to studying the effect of short-chain alcohols and endogenous peptides on lipid membranes. It has been shown that the addition of alcohols reduces the heart rate and mitochondrial ATP synthesis in Daphnia. On the other hand, when alcohols were added in combination with melittin, they increased both heart rate and ATP synthesis. Analysis of bilayers mimicking the inner mitochondrial membrane in terms of lipid composition showed that alcohols facilitate the adsorption of melittin on the membrane. After binding, the peptide induces the formation of non-bilayer structures that are important for the functioning of ATP synthase.
The manuscript is written clearly and presented in a well-structured manner. The citations are relevant. The presented results are well illustrated by figures and tables. The conclusions are confirmed by the obtained data. I reccomend to accept the manuscript «as is»
Author Response
We thank the reviewer's comments and the evaluation of our manuscript.
Reviewer 2 Report
The presented manuscript entitled „Short-chained Alcohols Make Membrane Surface Conducive for Melittin Action: Implication for Physiological Role of Alcohols in Cells” discusses the influence of of short-chain alcohols and membrane-active peptides – melittin as well as their synergistic effects on the Lipid packing and polymorphism of model liposomal membranes and correlates them with the experiments in Daphnia pulex. The manuscript is well and clear written and, so only some minor issues need to be corrected or explained.
1. In Materials and Methods p. 2.5 Statistics authors mentioned about using the Student’s t-test for analysing statistical significant of the results. Unfortunately, the results of this analysis have not been presented in the manuscript.
2. The part 3.4 Isolated and Combined Effect… lines 415-417, The authors discus the effect of alcohols and melittin on the permeability and structure of unilamellar liposomes. Author discus results presented in fig. 6 and table 1 as well as some not presented results. It is somewhat messy part of the manuscript. Please specify more precisely what system was investigated. For the results obtained for higher concentration of melittin (ratio 10:1) please just add additional rows to the table 1 to show the mentioned but not presented data. Also, why in fig. 6e, as exemplary results of the influence of the high concentration of melittin on the liposomal membranes, the data for solution of high concentrated melittin with ethanol (or both, melittin and melittin with ethanol? It is not clear. lines) is presented but in table 1 (last column) for the high concentrated melittin only (if the reviewer figures it out correctly)?
3. The part 3.4 Isolated and Combined Effect… lines 431-436 &452-472 Later, in the discussion about the quantitative changes of the Ii/Io ratio how was calculated the computer-extrapolated area? Was the signals deconvoluted for all components to asset their contribution?
4. The part 3.4 Isolated and Combined Effect… line 483 Based in the data form table 1 the content of NB fraction in liposomes treated with high concentrated melittin (M*) should be ca. 37 %.
Author Response
Authors’ answers to the reviewer # 2 comments
Authors are greatly thankful to this reviewer for his/her careful review of our manuscript and for his/her considered comments which are made to improve the quality of our manuscript. We provide our answers to the reviewer’s comments below:
Comments and Suggestions for Authors
The presented manuscript entitled „Short-chained Alcohols Make Membrane Surface Conducive for Melittin Action: Implication for Physiological Role of Alcohols in Cells” discusses the influence of of short-chain alcohols and membrane-active peptides – melittin as well as their synergistic effects on the Lipid packing and polymorphism of model liposomal membranes and correlates them with the experiments in Daphnia pulex. The manuscript is well and clear written and, so only some minor issues need to be corrected or explained.
Note to the reviewer: when looking for the lines indicated in Authors answers in the revised manuscript, keep “All Markup” on in Review.
- In Materials and Methods p. 2.5 Statistics authors mentioned about using the Student’s t-test for analysing statistical significant of the results. Unfortunately, the results of this analysis have not been presented in the manuscript.
Authors answer: As per the reviewer’s request, we present the analysis for statistical significance of the results in the Appendix in Tables A1 to A6.
- The part 3.4 Isolated and Combined Effect… lines 415-417, The authors discus the effect of alcohols and melittin on the permeability and structure of unilamellar liposomes. Author discus results presented in fig. 6 and table 1 as well as some not presented results. It is somewhat messy part of the manuscript. Please specify more precisely what system was investigated. For the results obtained for higher concentration of melittin (ratio 10:1) please just add additional rows to the table 1 to show the mentioned but not presented data. Also, why in fig. 6e, as exemplary results of the influence of the high concentration of melittin on the liposomal membranes, the data for solution of high concentrated melittin with ethanol (or both, melittin and melittin with ethanol? It is not clear. lines) is presented but in table 1 (last column) for the high concentrated melittin only (if the reviewer figures it out correctly)?
Authors answer: We fully agree with the reviewer that presenting in Fig 6E the1H-NMR spectrum of liposomes treated with high concentration of melittin + ethanol is not consistent with Table 1 where data on liposomes treated with high concentration of melittin in absence of melittin is presented. To correct this inconsistency and to address the reviewer’s recommendation we now present in Table 1 all data on membrane permeability and formation of non-bilayers both for low (1.2 ×10–4 M) and high (1.2 ×10–3 M) concentrations of melittin in absence and presence of all alcohols studied in this paper. We also extensively edited the content of the part 3.4 to make it easier to the readers to follow the narrative.
- The part 3.4 Isolated and Combined Effect… lines 431-436 &452-472 Later, in the discussion about the quantitative changes of the Ii/Io ratio how was calculated the computer-extrapolated area? Was the signals deconvoluted for all components to asset their contribution?
Authors answer: The signals were deconvoluted for Io, Ii and non-bilayer signals and areas of each signal were assessed. The permeability integers were obtained as Ii/Io, and the molar percentage of non-bilayer organized PC from the total moles of PC was assessed by calculating percentage of non-bilayer signal area from the area of entire spectrum including all its components: Io, Ii and non-bilayer signal (see lines 231-236 and 514-517 in the revised manuscript).
- The part 3.4 Isolated and Combined Effect… line 483 Based in the data form table 1 the content of NB fraction in liposomes treated with high concentrated melittin (M*) should be ca. 37 %.
Authors answer: We are thankful to the reviewer for spotting this typo. We corrected this typo on line 793 from 36 to 37 in the revised manuscript.
Reviewer 3 Report
In the current manuscript the authors investigated the effect of short chain alcohols on lipid membranes, and also the indirect effect on the interaction of the bee venom melittin. Here, melittin serves as a model peptide mimicking the action of endogenous membrane active peptides.
Since the action of short chain alcohols is still not well understood, the topic of the paper is a valuable contribution. It is also well conducted. The action of the alcohols and melittin on lipid membranes is addressed on different levels: effect on heart beat of daphnia, ATP synthesis of isolated mitochondria, order parameter and permeability of lipid membranes. The results consistently show that short chain alcohols increase the effect melittin exerts, with a u-shaped response curve: an increase of the strength of the effect with increasing number of carbon atoms for very short alcohols, followed by an increase starting roughly at 8 carbon atoms. The main effect of short chain alcohols seems to be to increase the tendency of the membrane to adopt non-bilayer structures, allowing better access of melittin to the lipids phosphate group.
However, prior to publication a few issues should be addressed:
1) Fig. 3: What about the spectra of ethanol alone?
and wouldn’t it be more appropriate to overlay spectra of the same type (perpendicular or parallel field) but with different composition? Then it would be easier to see the effect of the substances.
2) Line 355ff: effect of melittin on the order parameter S. In this paragraph, it is not clearly separated what is observation from the experiment, and what is interpretation.
The order parameter is derived from the change in the behaviour of 5-DSA (this is observed). Since 5-DSA is acidic, from its behaviour one can infer what happens to other acidic lipids (this is a (logical) assumption). An interaction of basic melittin with acidic 5-DSA can be expected. Why now can one infer that PC, a zwitterionic lipid, is affected in particular (line 357) ?.
3) Line 364: in case of PC it seems possible, that the alcohols allow better access of melittin to the phosphate groups. In case of acid phospholipids there is no such electrostatic repulsion…in particular of CL, which is the major acidic lipid in the mixture, the phosphate group is pretty much exposed anyway.
4) Line 545: The logical connection between the two parts of the sentence is unclear. The relation of the alkyl-chains of alcohol versus lipid bilayer is one aspect; the formation of micelles is another, which does not depend on the thickness of the bilayer.
5) Line 563: whether the requirement or more ATP requires a greater amount of non-bilayer structures, or the larger amount of non-bilayer is a by-product of the increased ATP consumption cannot be distinguished here, I would say.
6) Line 585ff: here the formation of pores at lipid to melittin ratios of 100:1 is described. Isn’t that in contradiction to a non-perturbed bilayer as found in the present study?
7) Line 637: concentrations around 100 µM are not low concentrations in case of melittin; many reported effects are observed at much lower concentrations. I assume, that in the present case the high concentrations required to observed an effect is, at least in case of the NMR and EPR experiments, a consequence of the very high lipid concentration.
Minor
1) line 46: “recreation substance”: maybe just “drug”?
2) line 48: not clear what the 10% refer to. 10% of what?
3) Line 66: why does the fact that the effect tapers off at longer chain length suggest that the short length alcohols affect the lipid phase?
4) Line 166: what is the molar fraction of 5-DSA?
5) Line 352: “At neutral pH, 5-DSA…”: what pH range does neutral pH include, and if outside this range: is then 5-DSA not useful anymore?
6) Line 382: it should be pointed out more clearly, that ferricyanide ions interact with PC only, because it interacts with the choline group
7) Line 395: this sentence is not entirely clear: is the new signal originating from PC/CL clusters, or from PC clusters, which are only formed if the membrane also contains CL?
8) Line 431 ff: the analysis procedure is not clear here. Wouldn’t it be possible to fit gaussians to the spectra to extract the contributions of Io, Ii and nb?
9) Line 464: in Tab. 1 are no data with lipid/melittin 10:1 in presence of alcohols..
10) Line 529: higher or lower than 70 to 1?
Author Response
Authors’ answers to the reviewer # 3 comments
Authors are greatly thankful to this reviewer for his/her careful review of our manuscript and for his/her considered comments which are made to improve the quality of our manuscript. We provide our answers to the reviewer’s comments below:
Comments and Suggestions for Authors
In the current manuscript the authors investigated the effect of short chain alcohols on lipid membranes, and also the indirect effect on the interaction of the bee venom melittin. Here, melittin serves as a model peptide mimicking the action of endogenous membrane active peptides.
Since the action of short chain alcohols is still not well understood, the topic of the paper is a valuable contribution. It is also well conducted. The action of the alcohols and melittin on lipid membranes is addressed on different levels: effect on heart beat of daphnia, ATP synthesis of isolated mitochondria, order parameter and permeability of lipid membranes. The results consistently show that short chain alcohols increase the effect melittin exerts, with a u-shaped response curve: an increase of the strength of the effect with increasing number of carbon atoms for very short alcohols, followed by an increase starting roughly at 8 carbon atoms. The main effect of short chain alcohols seems to be to increase the tendency of the membrane to adopt non-bilayer structures, allowing better access of melittin to the lipids phosphate group.
However, prior to publication a few issues should be addressed:
Note to the reviewer: when looking for the lines indicated in Authors answers in the revised manuscript, keep “All Markup” on in Review.
- 3: What about the spectra of ethanol alone?
and wouldn’t it be more appropriate to overlay spectra of the same type (perpendicular or parallel field) but with different composition? Then it would be easier to see the effect of the substances.
Authors answer: Effect of ethanol alone on the oriented EPR spectra was barely visible. We have now included description on effects of alcohols in absence of melittin on spectral anisotropy of the spin probe in lipid films (lines 341-349) in the revised manuscript. Spectral anisotropy of spin probes in well ordered lipid films is one of the strongest experimental pieces of evidence pointing to the bilayer packing of lipids in membranes. A decrease in spectral anisotropy manifested by gradual superposition of spectral lines obtained from parallelly and perpendicularly oriented spin probes offers dramatic visual evidence of polymorphic transitions in lipid phase. Therefore, we believe that presenting spectra obtained from only one type (parallel or perpendicular) orientation would be less appropriate in supporting the narrative of this paper.
- Line 355ff: effect of melittin on the order parameter S. In this paragraph, it is not clearly separated what is observation from the experiment, and what is interpretation.
The order parameter is derived from the change in the behaviour of 5-DSA (this is observed). Since 5-DSA is acidic, from its behaviour one can infer what happens to other acidic lipids (this is a (logical) assumption). An interaction of basic melittin with acidic 5-DSA can be expected. Why now can one infer that PC, a zwitterionic lipid, is affected in particular (line 357) ?.
Authors answer: We agree with the reviewer that the statement in line 357 with reference to PC is not a good statement, which is a result of poor editing on our side and is not what we wanted to state. We corrected this statement in lines 396-398 in the revised manuscript.
- Line 364: in case of PC it seems possible, that the alcohols allow better access of melittin to the phosphate groups. In case of acid phospholipids there is no such electrostatic repulsion…in particular of CL, which is the major acidic lipid in the mixture, the phosphate group is pretty much exposed anyway.
Authors answer: We totally agree with the reviewer that mentioning acidic phospholipids in line 365 is not consistent with the suggested details of electrostatic interaction between melittin and PC. We corrected this inconsistency in line 407 in the revised manuscript.
- Line 545: The logical connection between the two parts of the sentence is unclear. The relation of the alkyl-chains of alcohol versus lipid bilayer is one aspect; the formation of micelles is another, which does not depend on the thickness of the bilayer.
Authors answer: With an increase in length of alcohols’ alkyl chains, affinity of alcohols to membrane increases. At the same time the tendency of alcohols to form micelles also increases with increase in alcohols’ alkyl chains. At the cutoff alkyl chain length of alcohols (between 8 to 14 carbon atoms) more energy is released when alcohols form micelles outside membrane than when alcohols are inserted into membrane tot affect membrane structure. So, at the cutoff alkyl chain length, alcohols form micelles and do not affect the membrane structure. The cutoff effect of the alcohols’ alkyl chain length is well documented experimentally.
- Line 563: whether the requirement or more ATP requires a greater amount of non-bilayer structures, or the larger amount of non-bilayer is a by-product of the increased ATP consumption cannot be distinguished here, I would say.
Authors answer: The reviewer refers here to the sentence: “Obviously, to increase the heartbeat rate, more ATP must be produced, which – as inferred from data in [30] – requires greater amount of non-bilayer structures in the IMM.” In our view, this sentence clearly states that for more ATP to be produced greater amount of non-bilayer structures in the IMM is required.
- Line 585ff: here the formation of pores at lipid to melittin ratios of 100:1 is described. Isn’t that in contradiction to a non-perturbed bilayer as found in the present study?
Authors answer: Here the reviewer refers to the mechanism of pores formation and translocation of melittin through the pure PC membrane. Formation of pores that facilitates melittin translocation across the membrane is driven to release an asymmetrical intermembrane surface tension which does not lead to significant perturbance of membrane that would trigger formation of non-bilayer phase. In our present study the model membrane we used is made of 40 mol% of PC and the rest 60% includes PE, CL, PI and PS. At lipid to melittin molar ratio 100:1 our EPR study showed that melittin in absence of alcohols disturbs the surface area of membrane deep to the fifth carbon atom of alkyl chains which does not trigger formation of non-bilayer structures. Our 1H-NMR examination in this study also did not reveal formation of non-bilayer structures at lipid to melittin molar ratio 100:1 in absence of alcohols. Thus, we do not see that the previously reported results, at which the reviewer points in line 585ff (line 951ff in the revised manuscript), contradict to the finding of our present study.
- Line 637: concentrations around 100 µM are not low concentrations in case of melittin; many reported effects are observed at much lower concentrations. I assume, that in the present case the high concentrations required to observed an effect is, at least in case of the NMR and EPR experiments, a consequence of the very high lipid concentration.
Authors answer: In samples of mitochondria, lipid films and sonicated liposomes in this study the lipid to melittin molar ratio was 100:1 which is according to the studies of membrane-active peptides in lipid-peptide systems free of lipase activity is considered as a low melittin concentration. Melittin concentration of 1.65 × 10–4 M in samples of daphnia in this study is also low taking into account that daphnia samples were free of lipase activity, which would significantly activate membrane-lytic activity of melittin, and that only a portion of melittin molecules passes through the multiple barriers prior to reaching daphnia’s heart.
Minor
- line 46: “recreation substance”: maybe just “drug”?
Authors answer: As per the reviewer’s suggestion, we changed “recreation substance’ to “drug” (line 46 in the revised manuscript).
- line 48: not clear what the 10% refer to. 10% of what?
Authors answer: We clarified the question raised by the reviewer by adding “10% of ethanol inflicted diseases” (line 48 in the revised manuscript).
- Line 66: why does the fact that the effect tapers off at longer chain length suggest that the short length alcohols affect the lipid phase?
Authors answer: This reasonable comment of the reviewer is addressed by us in lines 66-70 in the revised manuscript.
- Line 166: what is the molar fraction of 5-DSA?
Authors answer: Molar ratio lipid to 5-DSA is 100:1 – see lines 186-187 and 332 in the revised manuscript.
- Line 352: “At neutral pH, 5-DSA…”: what pH range does neutral pH include, and if outside this range: is then 5-DSA not useful anymore?
Authors answer: 5-DSA is a DOXYL-stearic acid. In our present study it has been hydrated in buffer with pH 7.5. Although the charge on carboxyl group of 5-DAS is affected by the change in pH, paramagnetic properties of doxyl group of 5-DSA, which contains free radical, are not affected in the pH range 6 to 8 and according to some reports even outside this pH range. Thus, 5-DSA can be used in a capacity of spin probe in the pH range 6 to 8.
- Line 382: it should be pointed out more clearly, that ferricyanide ions interact with PC only, because it interacts with the choline group
Authors answer: We agree with the reviewer, and we addressed this point in lines 421-422 in the revised manuscript.
- Line 395: this sentence is not entirely clear: is the new signal originating from PC/CL clusters, or from PC clusters, which are only formed if the membrane also contains CL?
Authors answer: The new high-field signal is from choline group of PC in non-bilayer clusters that apart from PC also contain CL. We have clarified this point in line 442-446 in the revised manuscript.
- Line 431 ff: the analysis procedure is not clear here. Wouldn’t it be possible to fit gaussians to the spectra to extract the contributions of Io, Ii and nb?
Authors answer: We gave more details on the analysis procedure to quantitatively assess the membrane permeability (Ii/Io) and percentage of non-bilayer organized PC in liposomes in lines 231-236 and 522-525 in the revised manuscript.
- Line 464: in Tab. 1 are no data with lipid/melittin 10:1 in presence of alcohols..
Authors answer: As per the reviewer’s comment, we have now added in Table 1 data with lipid/melittin molar ratio 10:1 (Melittin concentration 1.2 ×10–3 M) in the revised manuscript – see line 533.
10) Line 529: higher or lower than 70 to 1?
Authors answer: We thankful to the reviewer for spotting this mistake. It should read lower. We now give the exact lipid to melittin molar ratio 67 to 1 in the revised manuscript (line 894).